# Attentional Flexibility Predicts A-Not-B Task Performance in 14-Month-Old-Infants: A Head-Mounted Eye Tracking Study

**DOI:** 10.3390/brainsci10050279

**Published:** 2020-05-05

**Authors:** Hanna Mulder, Carolien A. Van Houdt, Ineke J. M. Van der Ham, Stefan Van der Stigchel, Ora Oudgenoeg-Paz

**Affiliations:** 1Department of Pedagogical and Educational Sciences, Utrecht University, 3584CS Utrecht, The Netherlands; c.a.vanhoudt@uu.nl (C.A.V.H.); o.oudgenoeg@uu.nl (O.O.-P.); 2Department of Health, Medical, and Neuropsychology, Leiden University, 2333AK Leiden, The Netherlands; c.j.m.van.der.ham@fsw.leidenuniv.nl; 3Department of Experimental Psychology, Utrecht University, 3584CS Utrecht, The Netherlands; S.vanderStigchel@uu.nl

**Keywords:** looking behavior, infancy, executive function, attention, head-mounted eye tracking, A-not-B task

## Abstract

Early individual differences in executive functions (EFs) are predictive of a range of developmental outcomes. However, despite the importance of EFs, little is known about the processes underlying these early individual differences. Therefore, we investigated the association between 14-month-old infants’ attention on a reaching version of the A-not-B task and task success. We hypothesized that both strategic focused attention (measured as percentage looking time towards the correct location during delay) and attentional flexibility (measured as number of looks per second to available stimuli during delay) would relate positively to task performance. Infants performed the A-not-B task wearing a head-mounted eye tracker (*N* = 24). Results were trial-dependent and partially supported the hypotheses: (1) infants who were better able to flexibly shift attention between available stimuli on the first pre-switch trial showed better task performance overall; and (2) strategic focused attention to the hiding location during the first switch trial was positively related to performance on that particular trial only (trend-level effect). Thus, the study shows preliminary evidence that particularly attentional flexibility is a key factor underlying EF performance in young children. Advantages and challenges of working with head-mounted eye tracking in infants are discussed.

## 1. Introduction

Executive function (EF) is an umbrella term that is typically used to describe a number of higher-order cognitive processes needed for goal-directed, top-down behavior, and includes inhibition, working memory, and cognitive flexibility [1]. Over the past decade, interest in the development of EF has increased, because EF is a strong predictor of numerous child-related outcomes, such as academic achievement and behavioral regulation. Measures of EF have for example been shown to predict math scores in school aged children [2] and deficits in EFs relate to reading disabilities across childhood and adulthood [3,4]. Even very early in childhood, in the toddler and preschool years, EF abilities have been shown to be predictive of (pre-)academic skills and behavior problems [4,5].

Despite the importance of EF for a broad range of developmental outcomes, little is known about the processes underlying early individual differences in EF task performance: Why do some children perform better than others? Do different individuals focus on different (visual) information to facilitate EF task performance? Such information is important, as it may drive theory formation about EF development and ultimately serve to inform intervention efforts. Therefore, to improve our understanding of the nature of early individual differences in EF, the current study investigated the processes underlying EF task performance at a time when EF skills have only just begun to emerge. That is, we studied what infants, who are successful on an EF task, attend to while performing the task that distinguishes them from infants who are less successful on that task. In particular, the goal of the current study was to investigate which individual differences in 14-month-old infants’ visual attention on a reaching version of the A-not-B task are predictive of task performance. The A-not-B task [6] is considered a hallmark task of infant EF in the first and second years of life [7,8,9,10,11].

Research to date shows somewhat contrasting findings regarding the association between attention and EF task performance in infants and toddlers. First, Rivière and Brisson [12] showed that attentional flexibility was positively related to EF task performance in toddlers. In this study, 2.5-year-old toddlers performed a manual search task in which a toy was hidden under one of three boxes. Before infants were allowed to search for the toy, the boxes were moved to change locations in a fixed four-step order. Crucially, the box containing the toy was moved twice, interspersed by the moving of one of the empty boxes. Results showed that toddlers who were not successful in finding the toy looked at the box containing the toy significantly longer during its first move than toddlers who were successful in finding the toy. No other differences were observed in looking duration between the groups. Thus, the authors argued that some toddlers’ attention became “stuck” on the location of the box containing the toy after its first move, therefore making them more likely to fail the task. Moreover, a longitudinal study attests to the importance of attentional flexibility for EF task performance [13]. In this study, infants were classified as either “short lookers” or “long lookers”, based on their peak look duration towards a simple stimulus at age five months. Short lookers at five months obtained higher mean composite EF scores across a range of tasks administered between age two and four years than infants classified as long lookers. In addition, in a more recent longitudinal study, a composite measure of attentional flexibility at five months of age was shown to predict EF performance both at 10 months and at three years of age [14,15]. As such, in these studies, better attentional flexibility was related to more successful EF task performance, perhaps through more controlled or voluntary visual exploration of the task scene (see also [16]). 

Second, several studies have investigated infants’ gaze direction during the A-not-B task in relation to performance. In this task, infants have to retrieve a hidden object from one of two identical locations, following a short delay. Only after a number of consecutive correct retrievals at the same location (“A”) the object is hidden at the other location (“B”). In this way, infants build an automatic response to the A location due to several consecutive reaches to this location. Search performance after this switch of the hiding location from A to B taps into working memory and inhibitory control, i.e., keeping the hiding location in mind and inhibiting reaching to the previously correct location, respectively [8,17]. Despite directly observing how the object is hidden at B, young infants tend to perseverate and reach back to A (i.e., they make the A-not-B error).

Three studies have consistently shown that looking at the B location in the A-not-B switch trials is related to better task performance (i.e., successful search at B) in the second half of the first year of life [18,19,20]. For example, Horobin and Acredolo [20] showed that seven- to nine-month-old infants that solely looked at the B location during the delay phase outperformed infants that had a more mixed pattern of visual attentiveness (i.e., looking back to A, looking at the experimenter, or elsewhere). Based on their observations, the authors argued that infants appeared to be “employing a strategy of ‘keeping an eye’ on the target site” (p. 125). Recent work using eye tracking shows that 18-month-old infants tend to show more of such anticipatory looking to the correct B location than 10- and 12-month-old infants [21]. 

The finding that consistent looking towards the B location relates to better task performance potentially forms a contradiction to the observed negative association between “sticky attention” and performance on a search task observed by Rivière and Brisson [12]. Is keeping an eye on the correct hiding location an effective strategy or is it a sign of difficulty to disengage attention and lack of attentional flexibility? In other words, is focused attention or flexible attention related to optimal performance on a search task such as A-not-B? A recent study on attention and A-not-B task performance in five-month-old infants provides an initial answer to this question [22]. In this study, the proportion of time infants spent looking at a puppet stimulus and the duration of their peak fixation towards the puppet were measured as indices of focused attention and attentional flexibility, respectively. Although the two measures were significantly and strongly interrelated, both were unique predictors of A-not-B task performance, but in opposite ways: higher proportion looking time and lower peak fixation duration related to better A-not-B performance. The latter finding was restricted to children of mothers with low education. Thus, both focused attention and attentional flexibility appear to be important for infant performance on the A-not-B task. However, it is important to note that 95% of the infants in this study failed at very low levels of the adaptive A-not-B task that was used; these infants did not look for a toy within one or two buckets or failed at the pre-switch trials. Thus, performance of these young infants is more likely to reflect short-term memory and object permanence than EF. Therefore, it remains unclear how attentional focus and flexibility relate to EF performance as assessed with the A-not-B task. To investigate this, we tested how attentional focus and flexibility relate to A-not-B performance in older children, aged 14 months, when object permanence is well established and A-not-B performance is likely to reflect EF [23]. 

In the current study, we investigated how looking behavior predicts performance on a reaching version of the A-not-B task. Whereas previous studies have worked with live coding, video observations, or screen-based eye tracking to measure the direction of gaze of the infants [12,18,19,20,21], we used a relatively novel and precise technique to assess looking behavior towards 3D stimuli: head-mounted eye tracking (see [24]). Measures of attention were chosen to reflect both focused attention on the hiding location and attentional flexibility. Based on previous studies showing that looking at the correct location during delay could be an effective strategy facilitating performance on the A-not-B task [18,19,20], focused attention on the hiding location was operationalized as the percentage of time that infants spent looking at the correct hiding location during delay. Flexible attention was operationalized as the number of looks per second to the various regions of interest (ROIs; reflecting a pre-defined region the child could look at such as the hiding location, the distractor location, and the experimenter). The criterion for coding a single look to a particular region of interest was that a fixation of at least 100 ms took place and the infant was looking somewhere else both before and after that look. Thus, a larger number of looks is indicative of more switches between visual stimuli in general. 

To summarize, we investigated the association between focused and flexible attention during the A-not-B task, as measured using head-mounted eye tracking, and A-not-B test performance. A recent study shows that, despite being strongly interrelated, both processes—i.e., attentional focus and flexibility—uniquely contribute to A-not-B task performance in five months-old infants [22]. We investigated whether these findings extend to the beginning of the second year of life, when basic EF abilities have already emerged (e.g., [23,25]). We hypothesized that a larger percentage looking time to the correct location and a greater number of looks per second would positively relate to A-not-B task performance. In other words, infants who generally maintained focus on the hiding location interspersed with many brief looks away were expected to perform best on the task. Given that Riviere and Brisson [12]) showed that individual differences in visual attention occurring early on in an EF task were key predictors of task performance, we worked with the visual attention measures from the first few trials. Specifically, we tested how infant attentional state occurring just prior to the reaching response on the first switch trial predicted performance on that trial and on the task as a whole. 

## 2. Materials and Methods

### 2.1. Participants

Twenty-four infants were included in the current study. A further 35 infants were tested but refused to wear the eye-tracker equipment, and an additional 14 wore the equipment but their data could not be used for various reasons (child pushed off the eye tracker, *N* = 1; assessor procedural errors, *N* = 6; and difficulty calibrating the eye-tracker, *N* = 7). Thus, of the 73 infants taking part in the study, head-mounted eye tracking data of 24 infants were used for the analyses reported here (32.9%) (An additional four infants were given the A-not-B task with six instead of four hiding locations in the initial phase of the study; these infants are not reported on here).

Within the sample of 24 infants with head-mounted eye tracking data, age ranged from 14.3 to 14.9 months (*M* = 14.6 months; *SD* = 0.2) and 12/24 (50%) were boys. Most infants (23/24, 95.8%) had mothers with a college or university degree. For 19/24 (79.2%) infants, Dutch was the only language spoken at home. Parents reported that none of the infants were born preterm (gestational age < 37 weeks) or with low birth weight (<2500 g); were blind or deaf; or had brain damage, a chronic disease, or physical disability. There were no significant differences between these 24 infants and the remaining 49 infants in terms of age (*t*(70.6) = 0.205; *p* = 0.804), gender (χ^2^(1) = 0.552; *p* = 0.458), maternal education (χ^2^(1) = 0.003; *p* = 0.957), or home language (χ^2^(1) = 1.194; *p* = 0.275). 

### 2.2. Procedure

Infants were invited to the lab between 14 months and one week and 14 months and three weeks of age. When infants visited the lab with their parents, they were first given time to adjust to the lab environment and play with several toys, while the procedure was explained to the parents. Next, parents were asked for their written consent to participate in the study and to collect video recordings of the test session. The current study was part of a larger project, including multiple tasks. Only the A-not-B task was used in the current study. Tasks were always administered in a fixed order, and the A-not-B task was always administered first. During the A-not-B task, infants wore a Positive Science head-mounted eye-tracker (www.positivescience.com). The research reported in this article involves healthy human participants, and does not utilize any invasive techniques, substance administration, or psychological manipulations. Therefore, compliant with Dutch law, this study only required and received approval from our internal faculty board (Faculty’s Advisory Committee under the Medical Research (Human Subjects) Act (WMO Advisory Committee) at Utrecht University. Furthermore, this research was conducted, and written informed consent of each participant’s parent obtained, according to the principles expressed in the Declaration of Helsinki.

### 2.3. Head Mounted Eye-Tracker

The head-mounted eye-tracker consisted of a lightweight spandex cap on which headgear was strapped. The headgear consisted of two miniature cameras. One camera, the scene camera, faced outward to capture the field of view of the child. The other camera, the infrared eye-camera, faced inward and was adjusted so that it captured the child’s eye. An infrared LED was mounted next to the infrared eye-camera, which could be adjusted to illuminate the child’s eye to ensure dark-pupil tracking by creating a corneal reflection. Both the infrared eye-camera and LED were attached to an adjustable arm on the headgear. Infants also wore a vest to ensure safety while walking around, as in other tasks (not included in the current paper) children were allowed to move around the room. Infants either sat on their parents’ lap or on the ground next to the parent while the experimenter placed the eye-tracking equipment on them piece by piece (cap, vest, and headgear). Infants were distracted by both the parent and another experimenter with a video and several toys. They could also walk or crawl around if preferred. The scene camera’s field of view was 54.4° horizontal by 42.2° vertical. PSLive Capture V1.7.5 (Positive Science, New York, NY, USA) was used to record the videos of both the eye- and scene camera and the audio stream. Sampling frequency was approximately 30 Hz, as determined by the digital capture of the eye-camera.

*Calibration.* When infants had the full outfit on, they sat on their parents’ lap at the testing table for a procedure that allowed for offline calibration of the eye tracking data after the test session. The experimenter showed the infant a flat cardboard square with five windows for calibration (left and right, top and bottom, and middle, see Figure 1A). The cardboard square was held at approximately the same distance as the A-not-B test set-up was placed during the experiment, and the infant was seated at the same table and at the same height (i.e., on the parent’s lap) during calibration and testing. The calibration windows were approximately 5 cm by 5 cm each, surrounded by a 2-cm black border. The lowest windows were placed approximately 7 cm above the tabletop and the highest windows extended to approximately 27 cm above the tabletop. The windows to the left and right extended approximately 13 cm from the center of the calibration screen. Behind these windows, the experimenter rattled or squeaked a toy while watching through another window to see what the child was looking at. When the child looked at the toy behind one of the windows, the experimenter said “yes”. This verbal cue helped to accurately determine what the child was looking at in order to manually select these points during offline calibration. We built our calibration procedures on those of the group of researchers that co-developed the head-mounted eye tracker that we worked with [24,26,27]. Offline calibration was performed in Yarbus V2.2.8.1 (Positive Science, New York, NY, USA), using the five points of the cardboard square. 

### 2.4. A-Not-B Task 

After calibration, the experimental set-up of the A-not-B task was put in front of the child at a distance of 25 cm. The set-up in the current study consisted of four identical boxes with shutters, which the infants could open by pushing them inwards (see Figure 1B). The sizes of the boxes were 12 cm (width) by 30 cm (height), and the shutters covering the opening were 10 by 15 cm. The opening was located 10 cm above the tabletop and extended to 25 cm above the tabletop. This set-up was different from the set-up in a classical A-not-B task, as it had four rather than two search locations. Fourteen-month-old infants are already quite good at the classical A-not-B task, and making this task more difficult can be achieved by either a longer delay or more search locations and placing a screen between the child and the task display, among other options [17]. Since a longer delay was not practical, as this would give infants more time to touch or pull off the head-mounted eye tracker, the option of more search locations in combination with placing a screen between the child and the task display during delay was chosen. Locations 4 and 1, the two most outward locations, were used as Locations “A” and “B”, respectively. Two practice trials were conducted, in which the toy was partially hidden (Practice Trial 1) or completely hidden (Practice Trial 2) in Location “A”. 

During the test trials, the experimenter hid a toy in one of the locations (starting with Location “A”) and then put a cardboard screen in front of the hiding locations during a delay of one or three seconds. Experimenters were instructed to closely observe if the infant was watching the toy during the hiding procedure. If the infant was distracted during hiding, the experimenter first brought their attention back to the toy, and then repeated the hiding procedure. A video still of the infants’ view during hiding is shown in Figure 1C and an example video of an infants’ looking behavior during hiding is shown in the Appendix A. During the delay, the experimenter said “Wait a bit” (in Dutch, taking approximately one second) once for the one-second delay condition and three times for the three-second delay condition. After the delay, the cardboard screen was removed, and the experimenter asked the child where the toy was. The first box the child chose (i.e., the shutter the child touched first) was registered as the child’s response on that trial. Next, the toy was hidden at the same location again, until infants had searched correctly at that hiding location on two consecutive trials. After two consecutive correct searches, the toy was hidden at the other location (“B”). After each correct search, infants were allowed to play with the toy briefly, to ensure motivation to search was maintained throughout the task [8]. After an incorrect search, the experimenter showed the child where the toy was hidden and retrieved it from the hiding location.

The task consisted of two switch trials with a one-second delay, and two switch trials with a three-second delay, always given in the same order. Infants were given a maximum of four A trials before each switch trial. Thus, for infants who reached correctly on all trials, trial order was AA-BB-AA-BB, etc. For infants who erred only on the first A trial, trial order was AAA-BB-AA-BB, etc. When infants failed to search correctly at the hiding location on two consecutive A trials within the maximum of four attempts, task administration was terminated. Therefore, the number of switches infants could perform ranged from zero (if infants failed to search correctly at the hiding location on two consecutive A trials within the maximum of four trials before the first switch) to four. Experimenters were trained extensively to ensure reliable test administration. The number of switch trials that infants performed correctly was used as a measure of task performance (range zero to four).

### 2.5. Looking Behavior

Eye-tracking data of the A-not-B test trials were coded frame-by-frame using Datavyu (www.datavyu.org). For the purpose of the current study, looking behavior during the delay phase of the A-not-B task was coded. The delay phase started when the research assistant placed the cardboard screen between the child and the hiding locations (i.e., when the bottom of the cardboard screen was at the height of the top of the experimental set-up). The delay phase ended when the research assistant removed the cardboard screen (i.e., the cardboard screen was at the height of the top of the experimental set-up again), after which the child was allowed to search immediately.

During the delay phase, the screen in front of the experimental set-up prevented a good view of what hiding location the child was looking at exactly. However, approximately the top centimeter of the experimental set-up was still visible, with clear visual distinctions between the edges of each of the hiding locations (as they were separable boxes; see Figure 1B for the task set-up and Figure 1D for the infants’ view during delay). Because the child was sitting in front of the middle of the experimental set-up, the lighting and camera views provided clear information on which side (left or right) of the experimental set-up the child was looking at, while the distinction between the outer left (Locations 1 and 2) and outer right (Locations 3 and 4) two boxes was less clear. For this reason, we coded the left versus right side of the screen (in front of locations one and two versus locations three and four, respectively). Consequently, our measure of looking to the correct side may reflect looking to the correct hiding location or to the hiding location directly adjacent to the correct location. Similarly, looking to the wrong side may reflect looking to the previous hiding location or the hiding location directly adjacent to the previous location. As such, our measures of looking location are relatively crude proxy measures and need to be interpreted as such. Other task-relevant ROIs that we coded were the screen when it was not yet in front of the hiding location (beginning and end of delay phase) and the assessor. Finally, looking at any task-irrelevant location was coded as “elsewhere”. The coding scheme is provided in Appendix B. It should be noted that the ROIs reflect predefined areas in the experimental setup. The image a child sees in every single moment varies as children move their heads and torsos. This image might not include all ROIs and the size of the ROIs in children’s visual field vary as a result of their own movements. Therefore, data were coded manually frame-by-frame.

*Key variables.* Target variables for analysis were chosen to reflect strategic attentional focus on the hiding location and attentional flexibility. These variables were coded for the last A trial before the first switch (from here on, pre-switch trial) and the first switch trial (see Section 2.6 for further explanation about trial selection). As a measure of **focused attention**, we calculated the percentage looking time to the correct side of the display (referred to as “correct location” from here on) during delay by computing the sum of the duration of all looks to this ROI relative to delay duration. A single look was scored as a look to one of the ROIs with a duration of at least 100 ms [28]. In addition, we present descriptive statistics of percentage looking time to the two other most task-relevant ROIs, that is, the wrong side of the display (referred to as “wrong location” from here on) and the assessor. Missing eye tracking data were accounted for using the following formula:Percentage looking time to ROI = ((Sum of duration of all looks to ROI during delay)/(delay duration − missing duration)) × 100(1)

As a measure of **attentional flexibility**, the number of looks per second was used. For this measure, the sum of the total number of looks to each of the four task-relevant ROIs and the elsewhere category was computed relative to delay duration. The “elsewhere” category was included in this measure so that infants who frequently looked away from a test-relevant ROI and back again to the same location would obtain relatively higher attentional flexibility scores compared to infants whose attention remained on a particular task-relevant ROI consistently. This variable was computed as follows:Number of looks per second = ((Sum of number of looks to the four task-relevant ROIs and elsewhere category during delay)/(delay duration − missing duration in ms)) × 1000(2)

*Missing data.* Missing eye tracking data occurred occasionally when the point of gaze of the infant could not be established (for example, when the infant was blinking). Missing data were manually interpolated when missing duration was <150 ms, and the location of the last fixation before the missing was the same as the location as the first fixation after the missing (see [29]). Missing data with a duration longer than 150 ms were retained and accounted for in the proportion scores as described above. Trials with >50% missing eye tracking data were excluded from analyses.

*Reliability.* To investigate reliability of the looking behavior measures, 17% of trials (13/76 trials, including five A trials and eight B trials) were double coded by a second rater. Inter-rater correlations (Kendall’s tau) were acceptable to good for percentage looking time to the correct location (0.87; *p* = 0.001), percentage looking time to the wrong location (0.79; *p* = 0.002), and percentage looking time to the assessor (0.95; *p* < 0.001). The inter-rater correlation was suboptimal for the number of looks per second variable (0.64; *p* = 0.002) and percentage looking time to elsewhere (0.67; *p* = 0.011) and unacceptable for percentage looking time to the cardboard screen (0.44; *p* = 0.057).

Since one of our primary variables of interest, the attentional flexibility measure, did not fully pass common standards for inter-rater reliability, we explored whether reliability could be increased through working with more broad categorizations. A scatterplot showed that raters mainly disagreed on the exact coding of children with a relatively large number of looks per second. We therefore recoded the number of looks per second variables to two categories. To do so, we first checked whether the number of looks changed significantly from pre-switch to switch trial; this was not the case (*d* = 0.23; *p* = 0.440). We then computed the mean of the number of looks per second across the pre-switch and switch trial for each child and used the median of this variable to split the pre-switch and switch number of looks per second variables into two categories (i.e., below (0) and above (1) the median). Recoding resulted in 92% agreement and kappa 0.84 between raters, and categorical scores were used in the analyses.

### 2.6. Analyses

As previous work has shown that individual differences in visual attention occurring early on in an EF task may be key predictors of task performance [12], we focused our analyses on the visual attention measures obtained from the early phase of the task. Specifically, we selected visual attention measures that were most proximal to the first switch trial as indicators of infant attentional state in the early phase of the task: that is, looking behavior on the pre-switch trial just before the switch, and the switch trial itself. Another reason to work with visual attention measures from the early phase of the task only was that, in the adaptive A-not-B task protocol, not all infants progressed beyond the first switch trial, and looking behavior data were thus often missing for the later trials. A final reason for this selection of trials was that the aim of the study was to investigate how looking behavior predicted reaching behavior on the A-not-B task, rather than vice versa; by working with looking behavior data that were mostly generated *before* individual differences in reaching behavior occurred, the potential confounding influence of reaching behavior was accounted for. Note that the majority of infants (*N* = 18/24) passed the criterion of having two consecutive A trials correct at once, and thus followed the sequence A1[correct]-A2[correct]-B1. For these infants, visual attention data were used from A2 (pre-switch trial) and B1 (switch trial), respectively. A few infants (*N* = 6/24) needed more trials to pass criterion and followed sequences such as A1[fail]-A2[correct]-A3[correct]-B1 or A1[pass]-A2[fail]-A3[pass]-A4[pass]-B1. For these infants, visual attention data were used from A3 or A4 (pre-switch trial), respectively, and B1 (switch trial). To account for these relatively minor differences in reaching behavior, we ran our analyses in both the full sample and the subsample of infants who needed just two A trials to pass criterion and focused our interpretation on results that were consistent between these analyses.

To address the research questions, first, we studied looking behavior on the pre-switch and switch trial as predictors of performance on that switch trial using logistic regression. Second, we studied whether these looking behavior measures were predictive of performance on the total number of switches correct measure using linear regression. Given the small sample size, only univariate models were run and bootstrapping with 1000 resamples was applied. Given the non-normal distribution of the main outcome variable (number of switches correct) and some of the looking behavior measures, we used non-parametric testing for the descriptive statistics (Kendall’s tau correlations) and investigation of the impact of the number of A trials on looking behavior on the first switch trial (Mann–Whitney-U test). Analyses were conducted using IBM SPSS Statistics 22 (IBM, New York, NY, USA).

## 3. Results

### 3.1. Descriptive Statistics

#### 3.1.1. A-Not-B Task Performance (Reaching Response)

Children passed a mean of 1.3 out of four A-not-B trials successfully (*SD* = 1.2; range 0–4). Within the adaptive test protocol, 10/24 (42%) children passed the first switch trial (one-second delay), 13/24 (54%) passed the second switch trial (one-second delay), 5/24 (21%) passed the third switch trial (three-second delay), and 3/24 (13%) passed the fourth switch trial (three-second delay).

#### 3.1.2. Sample Size for Analysis

From the 24 infants, one did not reach criterion on the first set of A trials at all; thus, as this child did not have a valid “pre-switch” trial, the data were not included in any of the subsequent analyses. Further, data from one child with >50% missing eye tracker data on the switch trial were also excluded from the analyses of this trial. As such, the sample sizes were *N* = 23 and *N* = 22 for analyses pertaining to looking behavior on the pre-switch and switch trial, respectively.

#### 3.1.3. Delay Duration

Delay duration was on average 2483 ms (*SD* = 492 ms, range 1427–3501 ms) on the pre-switch trial and 2583 ms (*SD* = 605 ms, range 1529–4351 ms) on the switch trial. Delay duration on these trials was not related to the number of switches correct performance measure (Kendall’s tau: −0.19; *p* = 0.237 and −0.06; *p* = 0.700, respectively). Given the large variance in delay duration, however, we worked with looking behavior measures that were corrected for delay duration in the analyses (i.e., percentage looking time, number of looks per second).

#### 3.1.4. Looking Behavior

Descriptives of looking behavior data are shown in Table 1. Percentage looking time to the correct location decreased significantly from the pre-switch to the switch trial (*p* = 0.009), while percentage looking time to the wrong location increased (*p* = 0.033). Percentage looking time to the assessor did not change from pre-switch to switch (*p* = 0.496), nor did the number of looks per second (χ^2^(1) = 0.188; *p* = 0.665). Raw looking behavior data are shown in Figure 2 and Figure 3. Example video clips of an infant with high percentage looking time to the correct location during delay (high focused attention) and high number of looks per second during delay (high flexible attention) of the pre-switch trial are shown in the Appendix A (Appendix A, respectively). The study datafile used for analyses is available in the Appendix A.

It is of note that individual differences in looking behavior were generally large; for example, some infants did not look at the correct location at all on the switch trial, while others focused on this location nearly the whole delay time. Intra- and inter-trial correlations across the continuous looking behavior measures and task performance are shown in Table 2. Infants with higher attentional flexibility on the pre-switch trial were less likely to look at the correct location (A) on that trial than infants with lower attentional flexibility (Cohen’s *d* = 0.96; *p* = 0.019); no other significant associations occurred between attentional flexibility and the other looking behavior measures (*d*s −0.38 to 0.49; *p*s ≥ 0.292).

#### 3.1.5. Number of A Trials

Given that our main aim was to investigate how looking behavior predicts reaching behavior, rather than vice versa, we explored whether the number of A trials infants received in our adaptive test protocol was a potential confound in the analysis of looking behavior data. Specifically, we tested whether the number of A trials that infants received influenced looking behavior on the first switch (B) trial. Five of 23 infants (22%) required more than two A trials to pass the pre-switch criterion (two correct A trials in a row). Three of those infants required three A trials, and two infants required four A trials. We compared looking behavior on the first B (switch) trial between the five infants who needed more than two A trials to pass criterion and the remaining infants who needed just two A trials to pass criterion. Data for the group comparison are displayed in Figure 4. Percentage looking time to correct during the first switch trial was just over twice as large in infants who needed two A trials to pass criterion than in infants who needed more A trials (Cohen’s *d* = 0.68; *p* = 0.319). Infants in the latter group spent a larger percentage of time looking towards the assessor (*d* = 0.70; *p* = 0.164). No between-group differences were observed for percentage looking time to the wrong location (*d* = 0.10; *p* = 0.820). Although group differences in percentage looking time to the correct location and assessor were not significant, this was likely due to low power given the small sample of children requiring more A trials. Thus, we deemed the effect sizes large enough to consider the number of A trials a potential confounder. Therefore, we ran our main analyses in both the full sample and the subsample of children who passed the first two A trials, and focus on the results that were consistent between these analyses.

### 3.2. Main Analyses

#### 3.2.1. Predicting Performance on the First Switch Trial (Using Logistic Regression)

Percentage looking time to the correct location and number of looks per second on the pre-switch and switch trial were entered as predictors in univariate logistic regression, with reaching performance on the first switch trial as outcome (1 = pass; 0 = fail). Table 3 (Models 1a and 2a) shows that percentage looking time to the correct location on the pre-switch trial was not a significant predictor of reaching performance on the first switch trial. Percentage looking time to the correct location on the first switch trial was a trend-level significant predictor of reaching performance on that trial. Children who spent more time looking towards the correct location had a higher chance of passing that trial (i.e., reaching correctly; Figure 5). The 95% confidence intervals in the model in which the number of looks per second was entered as predictor of switch trial reaching performance (Models 1b and 2b) were too large to interpret, most likely due to the small sample.

#### 3.2.2. Predicting the Total Number of Switches Correct (Using Linear Regression)

Percentage looking time to the correct location and number of looks per second on the pre-switch and switch trial were entered as predictors of task performance (reaching measure; total number of switches correct) in univariate linear regression. Only the number of looks per second on the pre-switch trial was a consistent positive and significant predictor of reaching performance (Table 4, Model 3b; Figure 6). Thus, children who had more flexible attention just before the first switch trial (as indexed by more looks per second) showed a greater level of flexibility in their subsequent manual responses (as indicated by these children being more likely to pass a greater number of switch trials in total). (The sample size for analysis was *N* = 23 for pre-switch IVs and *N* = 22 for switch IVs, given that one child had missing eye tracking data on the switch trial. When this child was also excluded from the analyses with pre-switch IVs (*N* = 22), results were comparable to those shown in Table 3 and Table 4 (ORs did not change, *β*s changed by no more than 0.01)). Note that our measure of focused attention included also looks to the hand of the experimenter holding the hiding screen in front of the correct location (see also Figure 1D) and looks to a part of the child him/herself in front of the hiding location (i.e., the hand). When these looks were excluded from the focused attention measure, results were comparable to those shown in Table 3 and Table 4 (ORs changed by no more than 0.01, *β*s changed by no more than 0.02).

## 4. Discussion

The current study aimed to investigate how individual differences in attention relate to EF task performance early in the second year of life, through studying the relationship between looking behavior and performance on the A-not-B task. We combined a reaching version of the A-not-B task, using real 3D stimuli, with measures of visual attention by adopting a head-mounted eye tracker. We predicted that both strategic focused attention, as indexed by a larger percentage looking time towards the correct location during delay (“keeping an eye out for the target” strategy; see [20]), and flexible attention, as indexed by a greater number of looks per second, would be positively related to test performance (see also Marcovitch et al., 2016 [22]). Results show partial support for this hypothesis. The key findings are that attentional flexibility during the pre-switch trial was related positively to performance on the task as a whole, while strategic focused attention to the hiding location during the first switch trial was positively related to performance on that trial only. Many previous studies have investigated the role of attention to the hiding locations in reaching performance on the A-not-B task [18,19,20,30]; the novelty of the current work lies in the joint investigation of focused and flexible attention as measured during the task. To the best of our knowledge, only one previous study adopted a similar analysis, in younger children and with visual attention measures obtained from another task rather than the A-not-B task itself [22].

The finding that attentional flexibility on the pre-switch trial positively related to EF task performance aligns with results from a previous study in toddlers. Rivière and Brisson [12] showed that toddlers who were not successful on a manual search task spent more time looking at the correct hiding location at the outset of the experiment (during the first out of four moves of the boxes) than toddlers who were successful—that is, their attention remained “stuck”. When we consider the raw data of looking behavior on the pre-switch trial in Figure 2, it is evident that some infants focused only on the hiding location during delay of the pre-switch trial without a single look to another stimulus. This type of behavior resembles what has been described as “sticky fixation” in early infancy [31]. Previous studies use similar terminology (“sticky attention”) to describe looking behavior in toddlers [12] and preschoolers as well (for a review, see [32]). Hanania and Smith [32] suggested there may be “a protracted course in the development of selective attention—from nonselective attention, to attention that (…) appears “sticky”, to selective attention that is flexible” (p. 629). Individual differences in our data seem to align with this developmental model—with some infants displaying sticky attention and others already able to allocate attention more flexibly. Such individual differences in visual attentional flexibility predicted subsequent individual differences in EF task performance, with 24% explained variance.

At least one alternative explanation for the finding that increased attentional flexibility is related to better EF task performance needs to be considered. Figure 6 shows that children with higher attentional flexibility focused less on the A location during the pre-switch trial than children with lower attentional flexibility, and this difference was significant. Potentially, the weaker encoding of A on the pre-switch trial led to less of a “pull” back to A during the switch trial, facilitating optimal switch trial performance. Previous studies have shown that the strength of encoding of A is a key factor predicting whether infants will make the A-not-B error or not [30]. Could it be that children with higher attentional flexibility were simply less focused on the task (“lack of attention”), therefore coincidentally looked less at A, and performed better on the task as a whole for that reason? The data do not seem to support this explanation. First, looking at A on the pre-switch trial did not predict performance on the subsequent switch trial in our data (Table 3, Model 1A), nor did it predict overall task performance in a consistent way (Table 4, Model 3A). Second, we would not expect children with higher attentional flexibility to perform better on the A-not-B task overall if lack of attention was an underlying mechanism, given that we worked with an adaptive protocol: children who struggled to focus attention on the task would then be expected to also comply less with the reaching demands of the task, and thus not persist with multiple switch trials. Instead, the data seem to suggest that increased attentional flexibility may lead to increased readiness to pick up and respond to task relevant changes (such as the switch from A to B), thus leading to better overall task performance. Future experimental work, manipulating the focus on A in conjunction with the distribution of attention across the rest of the test space, is needed to test these ideas further.

It is of note that these relevant individual differences in infant flexible attention occurred during the pre-switch phase, which was the least challenging part of the test. Put differently, attentional flexibility on the least challenging part of the test was a predictor of flexibility in manual responses on the more challenging part; and only the latter part is typically considered to measure “EF”. This finding aligns with other studies showing that individual differences in attentional flexibility and associated speed of processing on very simple tasks in early infancy are important predictors of EF performance later in life (toddlerhood [13] and early adulthood [33]).

Findings on the switch trial showed that a greater percentage of looking time to the correct location was related to more successful performance on that trial specifically, corroborating the idea that “keeping an eye out for the target” could be an effective strategy on the A-not-B task for infants [20]. This finding should be interpreted with caution given that it was only trend-level significant and did not extend to performance on the task as a whole. We hypothesized that infants who generally maintained focus on the hiding location as a strategy to reduce working memory load, while also efficiently and quickly shifting gaze about to be able to pick up and respond to relevant assessor cues, would perform well on the task. Although our findings were somewhat consistent with this prediction, they were trial-specific and thus did not support the idea that these processes operate in parallel to predict unique variance in EF task performance. This contrasts to the results from a previous study on five-month-olds in which both measures of attentional focus and flexibility predicted A-not-B performance [22]. In the latter study, measures of attentional focus and flexibility were operationalized differently from our study and derived from looking behavior towards a puppet stimulus. Specifically, higher attentional focus was indexed by a higher proportion look duration towards the puppet, and higher flexibility was indexed by shorter peak look duration towards the puppet. Further work is clearly required to investigate how focused attention on task-relevant stimuli develops in conjunction with optimal attentional flexibility in different tasks, situations, and age groups (see also [22]).

Several limitations of the current study need to be considered. First, a very large number of infants refused to wear the head-mounted eye tracker during the A-not-B task and the sample size for analysis was therefore small. Consequently, the current sample size only allowed finding large effects. Second, although no group differences on the demographics between infants who did and did not wear the head-mounted eye tracker were found, we cannot be certain whether other individual differences, for example in temperament, may have existed between infants who wore and infants who refused to wear the head-mounted eye tracker. Third, in the current study, a screen was put in front of the hiding locations during delay to make the task more difficult and to ensure that the experimenter did not have to actively distract the infants to prevent them from reaching to the hiding location prematurely, before the end of the delay. As such, we could investigate what visual information infants focused on naturally. The screen, however, made it difficult to score looking at each individual hiding location reliably, but the distinction between the left (locations one and two) versus right (locations three and four) side of the test set-up was clearly visible. Consequently, our measure of looking to the correct side may reflect looking to the correct hiding location or to the hiding location directly adjacent to the correct location. Similarly, looking to the wrong side may reflect looking to the previous hiding location or the hiding location directly adjacent to the previous location. Our looking behavior measures thus need to be interpreted as relatively crude proxy measures. Previous studies on the A-not-B task which have either worked with multiple hiding wells (seven wells [17]) or a sandbox with no discrete hiding locations [34] have shown that the exact location of infants’ and toddlers’ reaching errors is reflective of the strength of the conflict between A and B. That is, when the correct hiding location was B (switch trial), infants who erred on the task more often reached in between A and B than on the other side of B—suggesting the conflict between locations is reflected in a “pull” back to A after having seen the object being hidden at B. The stronger this pull is, the closer the child will reach to A (or even at A) [34]. Similarly, looking towards the general direction of A—even if just adjacent to A—is thus indicative of a stronger pull to A than B, and vice versa. Therefore, these previous studies suggest that our measure of looking towards the general direction of the correct hiding location is informative of the strength of the child’s memory for that location. Fourth, through using the head-mounted eye tracking system we obtained video data allowing exact (frame-by-frame) coding of when the delay of the A-not-B task began and ended for each trial. Despite strict training of research assistants on the standardized A-not-B protocol, the delay length varied quite significantly between different trials (see Figure 2 and Figure 3). We believe such deviations may often be part of live administration with infants, but often go unnoticed; specifically, we checked all videos to ensure assessors adhered to the test protocol and did not note any obvious fluctuations in delay length; however, when measured precisely, such differences did exist. Note that delay length did not relate to test performance and was accounted for in the analyses through working with proportion scores. Fifth, the calibration screen was a flat cardboard while the test set-up was curved, causing some potential differences between the infants’ view during the calibration and test phase. In addition, differences in distance between the infant and the calibration screen and the infant and the test set-up occurred due to the infant’s relative freedom of movement while seated on their parent’s lap. This is a common feature of head-mounted eye tracking studies (see [26,27]). In screen-based eye tracking studies, the infants’ position is strictly controlled over time as infants are often strapped in a car seat in front of the 2D screen, allowing near perfect alignment between calibration and testing. The advantage that head-mounted eye trackers have over screen-based eye trackers is that they allow coding infant gaze during movement and in interaction with real 3D objects. Clearly, this advantage comes at the cost of losing strict control over distance parameters and a strict alignment between calibration and testing.

Finally, it was challenging to obtain acceptable reliability on our measure of attentional flexibility. Categorizing the variable provided a solution to this issue, as agreement between coders became >90% after categorization. This was the first study from our research group in which we worked with the head-mounted eye tracker, and thus we conducted a careful post-hoc analysis of the differences in raters to identify the potential issues that may have contributed to the discrepancies in the number of looks between raters. First and most importantly, a discrepancy in determining the exact location between coders occurred occasionally when the child was looking at the edge of one ROI which was directly adjacent to another (for example, screen versus assessor, when the assessor was holding the screen, and left versus right side of the screen when the child was focusing close to the center of the screen). As we were keen to investigate fine-tuned measures of looking behavior during a traditional infant EF task, we did not adjust the set-up of the A-not-B task specifically for the purposes of working with the eye tracker. Most of our ROIs were close together as a consequence and this study shows that this was a challenging endeavor. More reliable coding of attentional flexibility in a fine-tuned manner may be achieved by adjusting the lab and set-up such that ROIs are at a larger distance from each other (e.g., a set of small brightly colored toys on a white table in a head-mounted eye tracking study by Yu and Smith [35]). In doing so, clearly, the set-up becomes less ecologically valid as visual stimuli are often not neatly spatially separated from one another in the real world. Second, delay duration was relatively short and we worked with only two trials per child (pre-switch and switch). Within each trial, a range of one to nine looks were identified, while the differences between raters ranged from zero to two looks for almost all double coded cases. A difference of only one look between raters already constituted a 0.5 SD difference in the number of looks per second variable. Thus, working with longer delays and/or more trials would have made the number of looks variable less dependent on the accuracy of coding a single (short) look and would therefore potentially have made the measure more reliable. However, we specifically opted not to work with long delays procedures, as this would provide infants with more time to push away the eye tracker which may negatively affect reliability too. In our study, infants were given the eye tracker to wear for the full duration of the A-not-B task, but a few moved the eye tracker halfway through the task and/or pushed off the headgear altogether. To summarize, establishing the right balance between building procedures that are lengthy enough to capture reliable measures, without giving infants much time to push or move the eye tracker, was a challenge in our study.

In contrast to the difficulties we encountered with coding the number of looks variable, we obtained sufficient reliability in coding the percentage of time infants spent looking at the correct hiding location (see also Corbetta, Guan, and Williams [36], who make a similar point). Incorporating dynamic error margins around ROIs as was done in a recent study may further enhance coding reliability [37]. When considering the issues mentioned above, working with head-mounted eye tracking provides the opportunity to collect data on visual attention with good spatial and temporal precision from infants in relatively natural settings (for example, as they walk across a walkway in the lab [37]).

## 5. Conclusions

To the best of our knowledge, the current study is the first to use head-mounted eye tracking to investigate the associations between looking behavior on a reaching version of the A-not-B task and performance on this specific task. This study provides preliminary evidence that attentional flexibility is a key factor underlying early individual differences in EF performance in 14-month-old infants. Infants who were better able to shift attention between available stimuli, as indexed by a larger number of looks per second, showed better task performance. In addition, we found a trend-level significant effect of strategic focused attention on the hiding location (“keeping an eye out for the target” strategy) on performance, although the effect was limited to a single switch trial and did not extend to performance on the task as a whole. Further work is needed to investigate how early individual differences in attentional flexibility, strategic focused attention, task difficulty, and environmental factors interact in the course of EF development. In addition, head-mounted eye tracking offers unique opportunities to measure what infants have in view. Further methodological advances are required to ultimately achieve detailed measures of looking behavior that move beyond look duration to focal stimuli as infants act in the real, cluttered, and often disorganized world.

## Figures and Tables

**Figure 1 brainsci-10-00279-f001:**
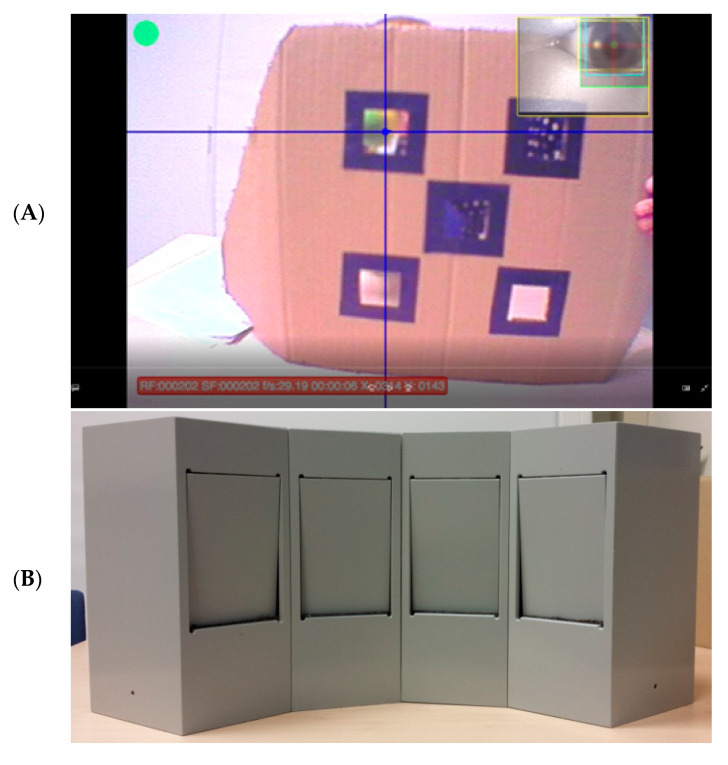
(**A**) Video still of the calibration screen, captured from the infant’s head-camera. The blue cross-hair marks the infant’s looking location. (**B**) A-not-B task with four hiding locations. (**C**) Video still of the A-not-B task set-up during the hiding procedure, from the infant’s head-camera. The blue cross-hair marks the infant’s looking location. (**D**) Video still of the A-not-B task set-up during delay, from the infant’s head-camera. The blue cross-hair marks the infant’s looking location. The blue arrow points to the distinction between the left and right side of the task display, which was used as a reference point during coding.

**Figure 2 brainsci-10-00279-f002:**
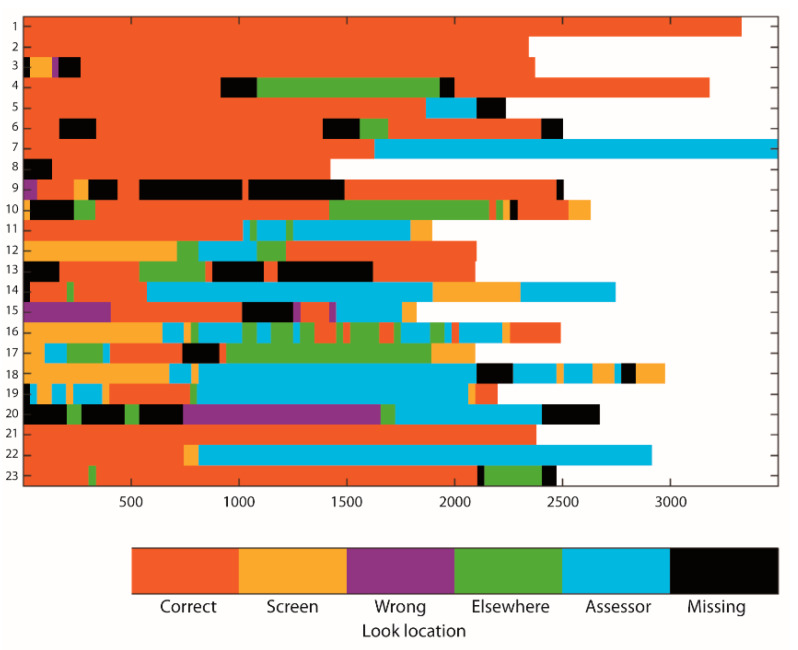
Raw looking behavior data (after interpolation of missing times < 150 ms) for the pre-switch trial. Children who passed the first two A trials only are Cases 1–18; children who needed more than two A trials are Cases 19–23.

**Figure 3 brainsci-10-00279-f003:**
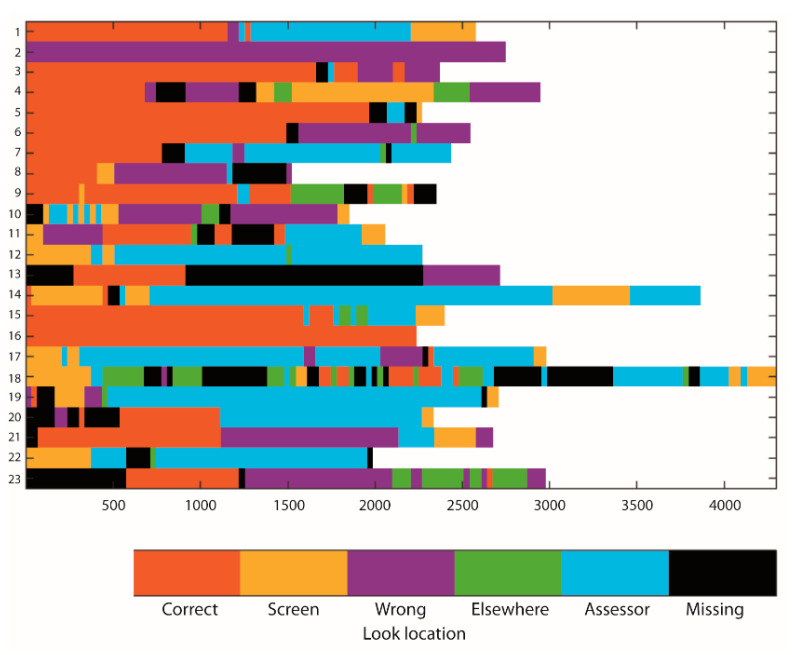
Raw looking behavior data (after interpolation of missing times < 150 ms) for the switch trial. Children who passed the first two A trials only are Cases 1–18; children who needed more than two A trials are Cases 19–23. Case 13 had more than 50% missing time and was therefore not included in the analyses.

**Figure 4 brainsci-10-00279-f004:**
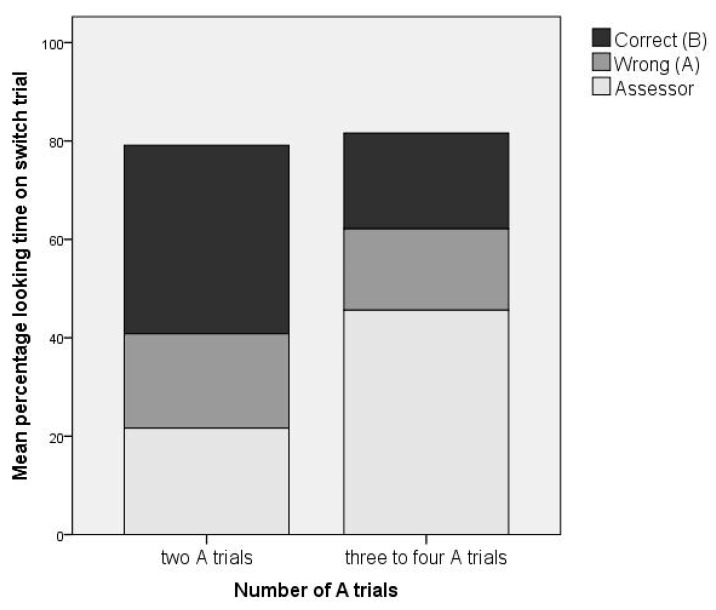
Percentage looking time to the correct location, wrong location, and assessor on the first (B) switch trial by the number of A trials needed to pass criterion (two A trials: *N* = 17; more than two A trials: *N* = 5).

**Figure 5 brainsci-10-00279-f005:**
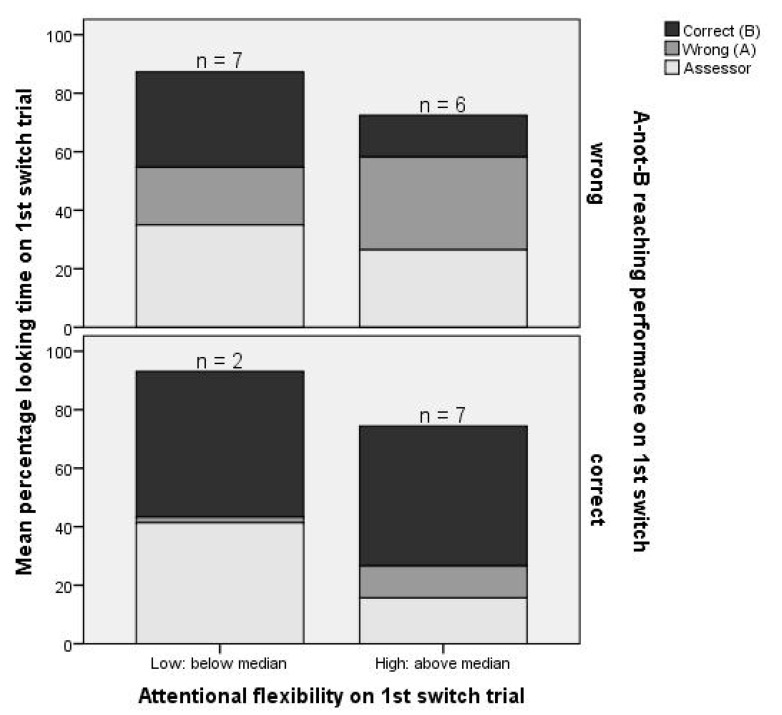
Reaching performance on the first switch trial in relation to looking behavior on that trial.

**Figure 6 brainsci-10-00279-f006:**
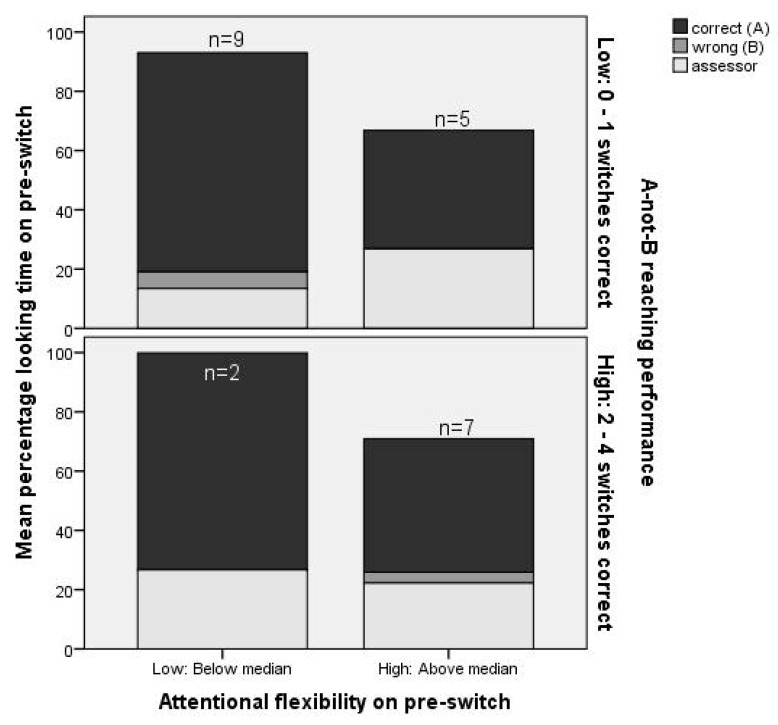
Attentional flexibility on the pre-switch trial in relation to the total number of switches correct (reaching response, dichotomized for presentation purposes) on the A-not-B task.

**Table 1 brainsci-10-00279-t001:** Descriptive statistics of looking behavior data (percentage looking time) on the pre-switch and switch trial (all children).

Trial	Pre-Switch Trial (*N* = 23)		Switch Trial (*N* = 22)	
	*M* (*SD*)	Range	*M* (*SD*)	Range
Correct	57.7% (35.2)	0.0–100.0%	34.0% (32.3)	0.0–99.6%
Wrong	3.3% (11.6)	0.0–50.7%	18.6% (26.9)	0.0–99.8%
Assessor	20.2% (26.0)	0.0–71.9%	27.1% (31.7)	0.0–82.8%

Percentage missing time on the pre-switch trial was 9.0 on average (*SD* = 12.8; range 0.0–43.1); percentage missing time on the switch trial was 8.0 on average (*SD* = 8.7; range 0.0–32.6). Data from one further child with >50% missing time on the switch trial were excluded.

**Table 2 brainsci-10-00279-t002:** Correlations (Kendall’s tau) between continuous looking behavior measures and the total number of switches correct performance measure (all children).

Trial	1	2	3	4	5	6	Performance
Pre-switch							
1. percentage looking time to correct		−0.27	−0.60 ***	0.23	0.43 *	−0.42 **	−0.23
2. percentage looking time to wrong		-	0.21	0.14	−0.26	0.14	0.21
3. percentage looking time to assessor			-	−0.22	−0.52 **	0.59 ***	0.17
Switch							
4. percentage looking time to correct				-	−0.18	−0.43 *	0.19
5. percentage looking time to wrong					-	−0.37 *	−0.13
6. percentage looking time to assessor							0.05

*** *p* < 0.001; ** *p* < 0.01; * *p* < 0.05.

**Table 3 brainsci-10-00279-t003:** Results of univariate logistic regression analyses predicting performance on the first switch trial by looking behavior.

	*χ*^2^ (*df*)	B (95% CI)	*SE*	OR (95% CI)
*All children*				
IV: Pre-switch trial variables (*N* = 23)				
Model 1a				
Percentage looking time to correct	0.330 (1)	−0.01 (−0.04 to 0.02)	0.03	0.99 (0.97 to 1.02)
Model 1b				
Number of looks per second ^a^				
IV: Switch trial variables (*N* = 22)				
Model 2a				
Percentage looking time to correct	3.206 (1) ^+^	0.03 (−0.006 to 0.009) ^+^	0.24	1.03 (1.00 to 1.06)
Model 2b				
Number of looks per second ^a^				
*Only children who passed the first two A trials*				
IV: Pre-switch trial variables (*N* = 18)				
Model 1a				
Percentage looking time to correct	1.312 (1)	−0.02 (−0.08 to 0.02)	0.93	0.98 (0.95 to 1.01)
Model 1b				
Number of looks per second ^a^				
IV: Switch trial analyses (*N* = 17)				
Model 2a				
Percentage looking time to correct	3.612 (1) ^+^	0.03 (0.003 to 0.15) *	2.60	1.03 (1.00 to 1.07)
Model 2b				
Number of looks per second ^a^				

^a^ Results not shown as confidence intervals around the OR were too large to interpret, due to the small sample size. B, 95% CI of B, *p*-values and *SE* based on bootstrapped results with 1000 resamples. The dependent variable is coded as 0 = fail, 1 = pass. ^+^
*p* < 0.10, * *p* < 0.05.

**Table 4 brainsci-10-00279-t004:** Results of univariate linear regression analyses predicting total number of switches correct.

	*R* ^2^	B (95% CI)	*SE*	*β*
*All children*				
IV: Pre-switch variables (*N* = 23)				
Model 3a	0.09			
Percentage looking time to correct		−0.01 (−0.02 to 0.001)	0.01	−0.30
Model 3b	0.24 *			
Number of looks per second		1.19 (0.32 to 2.02) *	0.43	0.49
IV: Switch variables (*N* = 22)				
Model 4a	0.06			
Percentage looking time to correct		0.01 (−0.01 to 0.03)	0.01	0.25
Model 4b	0.02			
Number of looks per second		0.35 (−0.63 to 1.37)	0.50	0.14
*Only children who passed the first two A trials*				
IV: Pre-switch variables (*N* = 18)				
Model 3a	0.24 *			
Percentage looking time to correct		−0.02 (−0.04 to −0.003) *	0.01	−0.49
Model 3b	0.25*			
Number of looks per second		1.29 (0.20 to 2.28) *	0.54	0.50
IV: Switch variables (*N* = 17)				
Model 4a	0.03			
Percentage looking time to correct		0.01 (−0.12 to 0.03)	0.01	0.17
Model 4b	0.05			
Number of looks per second		0.56 (−0.64 to 1.84)	0.65	0.21

B, 95% CI of B, *p*-values and *SE* based on bootstrapped results with 1000 resamples. * *p* < 0.05.

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
