# Peer review of "Attentional Flexibility Predicts A-Not-B Task Performance in 14-Month-Old-Infants: A Head-Mounted Eye Tracking Study"

_brainsci, 2020, doi:10.3390/brainsci10050279_

Round 1

Reviewer 1 Report

The authors did a commendable job addressing my concerns. 

Author Response

We would like to thank the reviewer for his/her positive response to the revision of our manuscript.

Reviewer 2 Report

The question raised by these researchers is interesting and their attempt to run this task using a head-mounted eye-tracker with 14mo infants to capture attention flexibility is quite commendable. Unfortunately, the manuscript, as a whole, is much difficult to read, it needs much clarifications, and the results are not very strong. Because head-mounted eye-tracking is extremely difficult to use with infants, it leads to high attrition rates and small sample sizes, which is a recurring general problem in the analyses presented. Here are my point-by-point more detailed comments.

Possible issues with parallax: The setup as shown in figure 1 is curved. The calibration is made with a cardboard (a flat cardboard held vertically I assume? This is no specified in the ms). Head-mounted eye-trackers are 2D systems that will provide most accuracy at the calibration window. As one looks away from the calibrated window, the accuracy of the point of regard degrades quickly, especially in the depth dimension as would be the case in this curved setup. Why did the researchers use a curved setup and not a straight one that would align perfectly with the calibration window?

The size of the setup and size of the calibration window are not provided. This is important. Accuracy is the greatest at the center of the window. The two most significant targets in the study are at the extremes of the setup so if the cardboard for calibration does not extend beyond the setup sides, there may be issues in tracking accuracy at the edges of the setup where the targets are located. Further, the researchers report in the discussion difficulties with coding points adjacent to 2 ROIs. With depth issues, difficulty coding adjacent points, and using targets away from the center of the calibrated area, I really wonder about the accuracy level and tracking validity confidence the researchers were able to obtain with their setup.

During hiding event in location A, was the child looking? That’s a critical point in A-not-B tasks, but not mentioned in this manuscript.

Why using only 2 consecutive trials prior to switch trial?

ROIs are poorly defined. What size are they? Where were they located on the occluding cardboard scene? How did the researchers determine that  these ROIs accurately defined the intended areas of interest when the coding was done during delay times and when only an occluding cardboard was seen on the scene? With head-mounted eye-trackers the scene changes and moves with the head, so the ROIs should move with the scene, but how can that be determined when the scene is a uniform cardboard?

Further, it is mentioned at some point that there are 5 ROIs, but they are never explicitly defined, and many results only refer to 3 looked areas (target A, target B, and assessor). Not sure what the other ROIs are, and if elsewhere is one of them.

There is mention in the method of two switch trials with 1 sec delay and two switch trials with 3 sec delay. Which switch trials have been used in the analyses?

P6, ln 240-244: The description of how the cardboard screen was placed relatively to the setup during delays is hard to understand.

The researchers never report how much active eye-tracking data they were able to use. They only say that trials with more than 50% data missing were not used. But percentages can do funny tricks at times. Say that I record 1 sec of active eye tracking in one 3 second delay switch trial. Fifty percent of active eye-tracking would correspond to 500 msec, but in this case 500msec could be misleading. Would 500 msec be enough to infer attentional flexiblity in 14-month-olds in a 3 sec delay trial? My inclination would be to say no, that’s not nearly enough. Reporting how much eye-tracking data was successfully collected and used is important to provide an idea of the corpus of data available for the analyses.

One look was defined as at least 100msec, but the researchers interpolated missing gaps in the data that were 150msec.

Median split for attentional flexibility should be better defined. I assumed above median split was more attentionally flexible, but this could be explicitly stated in the manuscript.

P7, ln 320: The majority of infants passed the criterion… How many was that?

Because the researchers measured looking and reaching to the targets, it is not always obvious which behavioral measure they used when they refer to the correct/wrong response. Be explicit in stating looking or manual response.

Why are the looks prior to reach considered the only predictors of response? What about the history of all prior reaches and prior looks? The history of prior reaches location has been shown to influence the success response rate in the switch trials. It is not clear why the analyses are limited only to the pre-switch and subsequent switch trials only?

P8, ln 349: analyses should be N=22 and matching for both pre-switch and switch trials. If the assumption is that pre-switch looking flexibility is important for performance on switch trials, then the child missing switch trial data should be eliminated from all analyses.

A lot of the descriptive data do not reveal trends due to the small sample size.

Why figure 4 data, on attentional flexibility, are not presented as a function of the motor response? To say that the above median group looks to the correct A target twice less (although not significantly differently) than the below median group, does not tell me if they were also more likely to switch correctly on the B trials. Without the motor data included, these figures are not very enlightening in regard to the authors’ hypotheses.

Same for the main analyses. The researchers show that the number of looks per seconds (figure 7, high or low) is the only variable that yielded significant differences in subsequent manual performance, but here it would have been useful to provide more info on where these more looks were directed to (location alone, figure 6, was not significant). Information provided in a piecemeal fashion does not provide the full picture. I think these patterns of how much and where infants looked should be used in a combined fashion to show how infant attention mounts to building the response on the switch trials.

Reviewer 3 Report

I appreciate the time the authors have taken to address the comments raised by myself and the other reviewer.

I am satisfied with the author’s response to my first comment about their use of 4 boxes as hiding locations but measuring looking to the left or right only. However, I am still not convinced by the authors arguments about the reliability of the looking data in the presence of the occluding screen. Where do children typically look at this age when an object that they were focussing on is hidden behind an occluder? There needs to be some discussion of the literature here. In the eye tracking data, is there evidence that the children are looking to the 1cm cue to the left and right side? The inclusion of figure 2 from the cover letter in the manuscript will be helpful to readers. When coding the data, how did you code looking data that fell on the boundary for the left and right ROI?

Round 2

Reviewer 2 Report

Thank you for providing more details about the method. I have now a better grasp of how the researchers ran their study, but new questions have arisen. Sorry for being picky, the points described below are quite significant.

I am still unconvinced about the calibration procedure and whether it allowed sufficient eye-tracking accuracy. It is hard to assess how the calibration frame shown in Figure 1a aligned with the A-not-B display in Figure 1b since the A-not-B display was not present during calibration (as show on Figure 1a). I had initially and logically assumed in my first reading of this work that the calibration frame was placed in front of the display for calibration, but figure 1a shows that this is clearly not the case. How did the researchers make sure that their calibration frame was properly aligned with where the display was going to be placed? Right now, if I compare figure 1b to figure 1a, figure 1a shows a poor alignment relatively to where the setup stood.

I appreciate the still frame on Figure 1c, but one frame is not very representative of the looking behavior as a whole. Can the authors provide a gaze plot (maybe hard to do with moving head), or even better a video of the eye-tracking sequence as supplementary material so that the readers can have a better idea of where the infants were looking during the hiding event?

Same for the figure 1d during delay, a video would be most useful, especially that this is the critical phase on which analyses were based on.

I am also quite concerned about this figure 1d showing the crosshair right on the experimenter’s hand who is holding the occluder. This study phase is critical to this study as it is used to assess how long and with what frequency of looks toddlers attend the area where the object was hidden, but this figure shows that the toddler looks at the experimenter’s hand, not the presupposed hiding location. This presents a significant confound to the researchers’ interpretation in my opinion as the researchers may not measure the focused attention they think they are measuring. If you present a plain board to infants, their looking pattern will naturally be attracted to where they can detect information – in this case, the hand of the experimenter. That is not a look or focused attention to where the object is hidden. In order words, infants may not display a look anticipation to where they saw the object disappear. The researchers may want to consider revisiting their analyses by removing any looks at the hand from their focused attention measure, and including only looks at the ROIs (this is going back to my previous comment on the importance of clearly defining ROIs). It seems that the ROIs, as I understand it now, were very broadly defined as right or left looks and may include looks that may not be related at all to the hiding location. Again, sharing a few randomly picked videos of where infants looked during the occluding time window would be extremely useful to understand where infants were looking during that period.

I also would recommend using a different pattern scheme for figure 3 than colors. It is nearly impossible to see the size of the different looking target categories on a B/W printer.

Finally, the researchers are wrong in their assumption that having freedom of movement helped cancel possible errors between calibrated area and tested area (discussion, p 21). It is exactly the opposite, more you allow freedom of movement, more you increase error in your eye-tracking measure relatively to the 2D calibrated area, inducing much parallax that distorts the location of point of regard. Based on the crosshair recording, it may appear that infants are looking at one location on the scene, while in fact they are looking off that particular location, but there is no way you can estimate the error or even detect it. That’s why calibration and how it is done is paramount to obtain valid eye-tracking recordings (hence all my questions). Eye-tracking systems that stand behind their strong promise of accuracy (like the EyeLink company for example) will ask researchers to immobilize the head of their participants as much as possible in context requiring precise eye-tracking, and certainly do not recommend providing greater freedom of movement. Eye-tracking accuracy of what is being measured in this study remains a strong concern.

There are some typos in the new legends.

Author Response

This manuscript is a resubmission of an earlier submission. The following is a list of the peer review reports and author responses from that submission.

Round 1

Reviewer 1 Report

The present paper investigates whether focussed attention and attentional flexibility during the A-not-B task predict task performance with 14 month old infants. The authors used head mounted eye tracking technology to measure looking behaviour to a live A-not-B task. Measures of looking behaviour were taken for the first switch trial and preceding pre-switch trial only. Looking to the correct location (focussed attention) during the delay for the first switch trial approached significance as a predictor of a correct search on the first switch trial. Number of looks per second (attentional flexibility) during the delay for the pre-switch trial predicted success on all subsequent switch trials.

This paper starts to address an interesting gap in our understanding of individual differences in executive functions. However, I have some concerns about the methodological design of the A-not-B task which was adapted for this study by including 4 boxes/hiding locations and a screen to occlude the boxes after the hiding event and before the search event.

While the authors clearly justify their use of 4 boxes/hiding locations, their measure of percentage of looking time to the ROI treats the task as the traditional two box task (left and right or correct and wrong) and as a result the looking data is potentially inaccurate. A child looking to the “correct” side might be looking in the direction of the incorrect box rather than the correct box. This significantly limits how the results can be interpreted.

The study uses looking behaviours to the correct and incorrect locations of a toy (one of four boxes) during the delay between the hiding and search event. However, during the delay a screen was moved into place to occlude the 4 boxes. It’s not clear how looking behaviour at the boxes can be measured when the boxes aren’t visible.

Reviewer 2 Report

The current study assessed focused attention and attentional flexibility in 14-month-old infants as they performed an A-not-B task. The study was novel in that: (a) this age range is older than typical for this task, (b) eye tracking was used to measure attention, and (c) attention was measured as part of the task during the delay period (as opposed to separate from the task). The authors found that their measure of attentional flexibility positively predicted performance, while focused attention to the correct hiding place predicted successful performance.

There is a lot to like about this study – the literature review is thorough (although I recommend including a reference to the meta-analyses performed on the A-not-B error: Marcovitch & Zelazo, 1999; Wellman, Cross, & Batsch, 1986), the study is well designed, and the findings are clear.

That being said, I have concerns about each major finding that reduces my enthusiasm to recommend publication.

(1) The measure of attentional flexibility is the number of different looks per second to an ROI. The logic, I assume, is that if there are many looks it implies that the infant is disengaging and reengaging often - a sign of flexibility. However, it also can be a sign of an inability to focus attention and is oddly reminiscent of attention deficiency as seen in slightly older children. This would explain why greater “flexibility” is leading to reduced correct looking on A trials. And, of course, if there is less attention to the A location during the A trials, then we would expect there to be less interference from A trials on the switch trial. Thus, better performance on the switch trial might be due to flexibility per se, or it might be an artifact of weaker encoding during the A trials.

(2) The finding that focused attention to the correct location increases correct searching is not novel, and is a central tenet of several theories including Diamond’s memory+inhibition account and dynamic systems theory. I am not convinced that replicating this in 14-month-olds and using advanced technology is a sufficient contribution. Interestingly, the authors themselves bring this up in the introduction.

There are also a number of minor issues that need to be addressed:

(1) In some places, the authors discuss the limitations of the small sample size. Should this study be published at all given this sample size?

(2) The primary analysis is rightfully an A-not-B task – that is to say an object is hidden several times at an A location and then hiding is switched to a B location. However, the continuation of this study where multiple switches happen can no longer be termed an ‘A-not-B’ task; rather, the term ‘delayed response’ task has been used in these cases.

(3) If the delay times were controlled to be either 1s or 3s, then why was there variability in the delay duration?